# Grain Refinement Mechanisms of TiC_0.5_N_0.5_ Nanoparticles in Aluminum

**DOI:** 10.3390/ma16031214

**Published:** 2023-01-31

**Authors:** Kui Wang, Haiyan Jiang, Qudong Wang, Yingxin Wang

**Affiliations:** 1National Engineering Research Center of Light Alloys, Net Forming, Shanghai Jiao Tong University, Shanghai 200240, China; 2State Key Laboratory of Metal Matrix Composite, Shanghai Jiao Tong University, Shanghai 200240, China

**Keywords:** nanoparticle, grain refinement, cooling rate, first-principles calculation

## Abstract

In this study, TiC_0.5_N_0.5_ nanoparticles (NPs) are shown to induce a remarkable grain refinement of aluminum at various cooling rates. The grain refinement mechanisms are systematically investigated by microstructure observation, edge-to-edge matching (E2EM) model prediction, and first-principles calculations. The experimental results suggest that as the cooling rates increase from 10 K/s to 70 K/s, a transition from intergranular to intragranular distribution of NPs occurs and the Al/TiC_0.5_N_0.5_ interface varies from incoherent to coherent. Based on the E2EM analysis combined with first-principles calculation, it is found that TiC_0.5_N_0.5_ can act as a potent nucleant for the heterogeneous nucleation of α-Al. By analyzing the NP effects on the nucleation and growth of α-Al, the grain growth restriction and nucleation promotion mechanisms are proposed to elucidate the refinement phenomena at low and high cooling conditions, respectively.

## 1. Introduction

Microstructure control is of particular importance in the metal casting industry, and it is often the case that a uniformly fined, equiaxed grain structure is favorable for both cast and wrought alloys, as it benefits the casting process and improves the mechanical properties of the cast metal, facilitating subsequent mechanical working [1,2,3]. Grain refinement by inoculation is a well-established practice, which is usually accompanied by the addition of a grain refiner to the metallic melt [4,5]. For effective grain refinement, not only do the nucleants in the grain refiner need to provide potent heterogeneous sites for nucleation of the primary solid phase, but also some solute elements are required in the melt to restrict the grain growth and facilitate nucleation, either at a columnar front competing with equiaxed solidification or from particles where nucleation has already occurred [6,7,8].

While grain refiners are widely used in the foundry, their performance is not always satisfactory. Research shows that the refiner effectiveness is influenced by many factors, including the size and size distribution of nucleating particles, the alloy composition, as well as the cooling rate [9,10,11]. Firstly, according to the free-growth model [9], the critical undercooling at which the particle can be activated for grain initiation is inversely proportional to particle size, and because of the particle size distribution, only a minority of the particles can reach the critical undercooling where they are active. Therefore, grain refinement often has low efficiency, with <1% of the particles acting as effective nucleants. Furthermore, the refiner effectiveness is dramatically compromised in the presence of certain elements. For example, the grain refinement via Al-Ti-B inoculation is achievable for a majority of Al alloys. However, it has a poor refining effect in Zr/Si/Cr-containing Al alloys. This is attributable mainly to the loss of nucleation potency by the formation of intermetallic phases with Zr/Si/Cr that possess a poor crystallographic matching with α-Al [12,13,14]. In addition, recent research has demonstrated that a deterioration in the grain refining effect occurs at increased cooling conditions, including high-pressure die casting [15]. Ali and Liang et al. [16,17] suggested that such a grain coarsening effect at high cooling rate results from the small constitutional undercooling zone ahead of the solid/liquid interface due to the high temperature gradient.

Most recently, the practice of incorporating nanoparticles (NPs) into the metallic melt has presented exciting grain refinement results for Al and Mg alloys [18,19,20]. For instance, J. Xu et al. achieved the control of architecture and microstructure of an Al composite by the assembly of NPs with liquid Al in molten salt [20]. Unlike the inoculation, the added NPs are capable of restricting the grain growth through their accumulation at the solid/liquid interface. The physical nature of NP-induced refinement makes it become a potential avenue to address the intrinsic limitations of inoculation. In previous studies [21,22,23], the NPs are taken to serve as growth restrictors rather than nucleants. However, if possessing high nucleation potency, NPs are likely to promote nucleation of the solid phase. Although the growth-restricting effect of NPs has been experimentally observed by synchrotron X-ray tomography [19,20], the exact mechanisms underlying the grain refinement by NPs remain incompletely understood, especially from the perspective of their effects as heterogeneous nucleants. First-principles calculation based on density functional theory (DFT) is taken as a powerful and efficient tool for the prediction and estimation of the nucleation potency of nucleants. Up to now, little experimental work has been conducted to clarify the nucleation potency of NPs for α-Al, especially by the DFT calculation. Moreover, the cooling rate is a crucial factor affecting the grain size of cast metals. It is well accepted that a high imposed cooling rate can produce a large thermal undercooling at which NPs may be activated for heterogeneous nucleation [9,16,17]. Hence, the purpose of the present study is to understand the combined effects of cooling rate and NP addition on the grain refinement of aluminum. Density functional theory (DFT) was employed to quantitatively analyze the nucleation potency of TiCN as the nucleant for α-Al. The research results may be conducive to further revealing the NP-induced refinement mechanisms.

## 2. Experimental Procedures

In this work, commercially pure (CP) Al ingots at 99.7% purity were firstly melted using an electrical resistance furnace. After complete melting and mixing of the raw metals, 0, 0.5, 1.0, 1.5, and 2.0 vol.% TiC_0.5_N_0.5_ NPs with the mean size of ~100 nm were added into the melt and dispersed ultrasonically at 720 °C for 15 min under Ar gas protection. Then, a sonotrode was taken away from the melt and it was reheated to 750 °C before the pour was completed in a preheated cast iron mold. The average cooling rate is determined to be around 3 K/s. To achieve various cooling rates, the melt was also poured into a V-shaped copper mold where the cooling rate changes from 10 to 250 K/s. For comparison with the conventional inoculation, 0.1, 0.3, and 0.5 wt.% Al-3B and Al-5Ti-1B (wt.%) grain refiners were also added to the Al melt, respectively, followed by stirring and 5 min holding, before casting in the preheated cast iron mold.

The specimens for microstructural observation were sectioned from the bottom of each casting and grounded and polished mechanically. The etched specimens were characterized by an optical microscope with polarized light. The grain size was determined via the linear intercept method, as described by ASTM E112-10. The microstructure was examined by JEOL 78600F SEM, JEOL 2100 TEM, and JEOL 2100F FEGTEM.

In order to analyze the atomic structure of the interface, first-principles density functional theory (DFT) calculations were conducted by Vienna Ab initio Simulation Package (VASP) [24] with the Projector Augmented Wave (PAW) [25] method, and the Perdew–Burke–Ernzerhof (PBE) variant of generalized gradient approximation (GGA) [26] was used. The plane wave cutoff energy was equal to 520 eV. The energy tolerance for the electronic relaxation was 10^−6^ eV. The Hellmann–Feynman force tolerance for the ion relaxation was 0.01 eV·Å^−1^. The Brillouin zone was sampled with a Gamma-centered *k*-point mesh. For the slab, a 12 × 12 × 1 *k*-point mesh was used and for the bulk a 12 × 12 × 12 *k*-point mesh was used. The Monte Carlo Special Quasi-random Structure (MCSQS) method [27] was employed for modeling the bulk TiC_0.5_N_0.5_ with the N-site in L1_2_ TiN randomly substituted by C. In order to avoid surface–surface interactions, a vacuum space of at least 10 Å was imposed in the supercell [28].

## 3. Results and Discussion

The typical microstructures of the castings are illustrated in Figure 1. It is evident that the coarse columnar grains dominate in the microstructure of the CP Al. After an initial addition of 0.5 vol.% TiC_0.5_N_0.5_ NPs, the coarse columnar grains show a pronounced shift towards the refined equiaxed grain structure, and the average grain size is decreased from 1350 to 285 μm. Further addition of NPs to 2.0 vol.% triggers a significant refinement of equiaxed grains, and the average grain size is further reduced to 138 μm. It is also notable that the grain size shows a sensitivity to the NP addition level, and no obvious saturation of refining effect is observed even at high additions, implying the high refinement efficiency of TiC_0.5_N_0.5_ nanoparticles.

Figure 2 shows the comparison of NP-induced grain refinement with the conventional inoculation. The macrostructure of the CP Al castings is characterized by coarse columnar grain structure. When the addition level is raised, a large quantity of equiaxed grains is formed at the center of the sample inoculated by the Al-3B mater alloy. However, as no titanium element can be used as the growth restrictor, the grain refinement effect is far from satisfactory. In stark contrast, a pronounced reduction in grain size and transition to equiaxed grain structure are observed throughout the sample inoculated by the Al-5Ti-1B master alloy. In addition, an increase in the addition level results in fine equiaxed grain structure. As compared to inoculation, a stronger grain refinement occurs when TiC_0.5_N_0.5_ nanoparticles are incorporated into the melt. The finer and more uniform equiaxed grain structure shown in Figure 2 demonstrates that TiC_0.5_N_0.5_ nanoparticles can induce a more effective grain refinement than conventional inoculation.

To unveil the influence of cooling rate on the NP-induced microstructural evolution, a wedge chill casting technique [29] was employed to produce the rapidly solidified specimens, which were subjected to various cooling rates. The anodized micrographs from the different positions on the V-shaped specimens with different NP contents are presented in Figure 3. Additionally, included in Figure 3 are the variations in grain size with cooling rate. It can be seen that the grains of the CP Al maintain the columnar structure, although an increased cooling rate leads to the finer α-Al grains. With the addition of 2 vol.% NPs, the coarse columnar grains are transformed to refined equiaxed ones, and the enhanced cooling rate leads to the further gain refinement. When the cooling rate reaches 70 K/s, the mean grain size is reduced up to 78 μm.

It is obvious that an increased cooling rate can lead to a decreased grain size by producing an increased melt undercooling. However, because no effective nucleants in the melt can induce the heterogeneous nucleation of α-Al, the bulky grains with columnar structure prevail in the matrix. By contrast, the samples with NP additions evince much more refined and homogeneous grain structure. This discrepancy can mainly be attributed to the NP influence on the nucleation and growth of α-Al. The microstructure evolution substantiates that the NPs might outweigh the cooling rate in terms of grain refinement.

Figure 4 illustrates the distribution of NPs in the matrix of the V-shaped sample with the local cooling rate of 12 K/s. It is evident from Figure 4a that a myriad of NPs with the mean size of ~100 nm assemble onto the grain boundary. The EDS analysis in Figure 4b confirms the presence of TiCN nanoparticles. In addition to discrete nanoparticles, many nanoparticles aggregate together to form nanoclusters at the grain boundary. As depicted in Figure 4b, the accumulation of these NPs along the grain boundary results in the formation of an NP layer in between the α-Al grains. Figure 4c shows a nanoparticle located at the grain boundary. From the crystallographic information provided in Figure 4d, this nanoparticle has a face-centered cubic structure with the unit cell of NaCl-type, which corresponds to TiC_0.5_N_0.5_ [30]. Figure 4d also highlights the interfacial structure between the α-Al and the intergranular NP. It is indicated that no fixed or preferential crystallographic orientation relationship (OR) can be observed and thus an incoherent interface is formed between them.

From the analysis above, it may be speculated that the grain refinement which occurs at relatively low cooling conditions is predominantly ascribed to the growth restriction caused by NPs. According to the nanoparticle capture model proposed by Xu et al. [31], NPs tend to be pushed by the solidification fronts because the solidification front velocity is usually much smaller than the critical velocity for nanoparticle capture during regular solidification processes for which the cooling rate is generally very low. As crystallites grow, the NP pushing by the solid/liquid interface occurs, and then the assembly of NPs onto the grain boundary gives rise to an NP layer to significantly suppress the solute atom migration, thus restricting the grain growth.

On the contrary, once the solidification cooling rate exceeds 70 K/s, a great majority of NPs are distributed inside α-Al grains, as illustrated in Figure 5a. Figure 5b reveals the intragranular distribution of NPs. Instead of accumulating at the growing interface, they are evenly distributed in the Al matrix. Figure 5c presents a nanoparticle embedded inside the α-Al grain. From the HRTEM analysis in Figure 5d, it can be seen that the TiC_0.5_N_0.5_/Al interface with little formed reactant is clear and smooth. Furthermore, TiC_0.5_N_0.5_ forms a coherent interface with α-Al. Hence, a fixed OR can be achieved and is ($11\bar{1}$)_Al_ parallel to ($11\bar{1}$)_NP_, and [011]_Al_ parallel to [011]_NP_. Figure 5e exhibits the other nanoparticle entrapped in the matrix. Likewise, it possesses a coherent interface with the α-Al as shown in Figure 5f, and the OR is given as ($1\bar{1}1$)_Al_ [$\bar{1}12$]_Al_//($1\bar{1}1$)_NP_ [$\bar{1}12$]_NP_.

To better understand the effect of TiC_0.5_N_0.5_ NPs on the grain refinement of α-Al at enhanced cooling conditions, their nucleation potency was examined in terms of the crystallographic matching, the work of adhesion, and the interfacial energies of Al/TiC_0.5_N_0.5_ interfaces.

As far as the crystallite nucleation is concerned, the edge-to-edge matching (E2EM) model proposed by Zhang et al. [32,33,34] is well documented to be effective in evaluating potential ORs between a nucleant substrate and solid phase. The interatomic spacing misfit (*f_r_*) and the interplanar spacing mismatch (*f_d_*) are the two pivotal factors in this model. If they are less than 10%, an OR between two phases can be predicted.

The possible orientation relationships between TiCN and Al are illustrated in Figure 6. Four low-energy and low-index interfaces between TiCN and Al are considered, including (111)_Al_//(111)_TiCN_ with Ti and C/N termination and (200)_Al_//(200)_TiCN_ and (220)_Al_//(220)_TiCN_ interfaces, in which the close-packed planes are matched across the interface. By calculation, seven potential ORs are given in Table 1. The determined values of *f_r_* and *f_d_* are less than 6.0% for almost all the potential Ors, implying a desirable crystallographic match. As commented by Zhang [32], when *f_d_* ≤ 6%, the matching planes are parallel or nearly parallel. The predicted Ors involving a plane pair of {1 1 1}_Al_//{1 1 1}_TiCN_ are consistent with the HRTEM analysis in Figure 5. The combination of experimental results and the E2EM analysis indicates that pairs of Ors and interfaces may exist between Al and TiC_0.5_N_0.5_. Consequently, α-Al may be able to nucleate and grow on the surface of TiC_0.5_N_0.5_.

Based on the Ors predicted by E2EM, three supercells containing (111)_Al_//(111)_TiCN_, (200)_Al_//(200)_TiCN_, and (220)_Al_//(220)_TiCN_ interfaces, respectively, were constructed to further reveal the nature of Al/TiC_0.5_N_0.5_ interfaces through density functional theory (DFT). The supercell models of (100), (110), and (111) Al/TiC_0.5_N_0.5_ are displayed in Figure 7a–c, respectively. The models of the polar Al_2_Ti_2_NC interface are the (1 1 1)_Ti2NC_ || (1 1 1)_Al_, (1 1 0)_Ti2NC_ || (1 1 0)_Al_, and (1 0 0)_Ti2NC_ || (1 0 0)_Al_, in which the close-packed planes are matched across the interface with N/C-terminated Ti_2_NC. To accommodate the periodic boundary condition inherent in a supercell calculation, we invoke the coherent interface approximation in which the softer Al matches the dimensions of the Ti_2_NC. To make the results comparable and reasonable, the ratio between the content of C and that of N is adopted as 1 on the interface for all Al/Ti_2_NC, and all the models are strictly stoichiometric, Al_2_Ti_2_NC.

The ideal work of adhesion (*W_ad_*), which is defined as the reversible work against the adhesion strength of interface atoms, is commonly used to estimate the interfacial quality [35]. It can be determined by the difference in total energy between the interface and its isolated slabs [35,36]:(1)$$W_{ad}=(E_{Al}^{Slab}+E_{TiCN}^{Slab}-E_{Al/TiCN}^{Slab})/S$$
where $E_{Al}^{Slab}$ and $E_{TiCN}^{Slab}$ are the total energy of the fully relaxed, isolated Al and TiCN slabs, respectively, and $E_{Al/TiCN}^{Slab}$ is the total energy of the Al/TiCN interface system. *S* is the total interface area of the unit cell.

Figure 7d shows the calculated *W_ad_* for the three interface structures. There is an obvious trend of increasing *W_ad_* with the index of the lattice plane. The Al/TiC_0.5_N_0.5_ (111) interface exhibits the largest *W_ad_* value of 6.6 J/m^2^, whereas the (100) interface shows the smallest *W_ad_* value of 1.2 J/m^2^. Increased *W_ad_* is correlated with increased thermodynamic stability. Therefore, the Al/TiC_0.5_N_0.5_ (111) interface is expected to be the most thermodynamically stable.

The interfacial energy is an important measure of interfacial stability, which can be defined as [35,36]
(2)$$\varepsilon =({Ε}_{Al/TiCN}-\frac{n_{1}}{{Ν}_{1}}{Ε}_{Al}^{Bulk}-\frac{n-n_{1}}{{Ν}_{2}}{Ε}_{TiCN}^{Bulk})/S$$
where *ɛ* is the interfacial energy, *n*_1_ and *n* are the number of Al and all atoms in the interfacial structure, respectively, $E_{Al}^{Bulk}$ and $E_{TiCN}^{Bulk}$ are the per-atom energy of the relaxed bulk Al and TiCN in the same supercell consisting of *N*_1_ and *N*_2_ atoms without vacuum, respectively, $E_{Al/TiCN}^{}$ is the total energy of the Al/TiCN interface system, and *S* is the total interface area of the unit cell. Generally, the lower the interfacial energy, the more stable the interface structure.

Figure 7e shows the interfacial energies of three Al/TiC_0.5_N_0.5_ interfaces. Although the matching plane pairs predicted by E2EM have the similar *f_d_* values, there is a great difference in interfacial energy. From Figure 7d, it can be seen that the interfacial energy is decreased with increasing the index of the lattice plane, and the Al/TiC_0.5_N_0.5_ (111) interface has the lowest interfacial energy of −11.7 J/m^2^. For all three Al/TiC_0.5_N_0.5_ interfaces, the calculated interfacial energies are far less than the solid–liquid interfacial energy of Al (0.158 J/m^2^) [9], which means that the α-Al grain is able to nucleate on the TiC_0.5_N_0.5_ surface.

Based on the DFT analysis, it can be inferred that of the three Al/TiC_0.5_N_0.5_ interfaces, the (111) interface with the largest work of adhesion and smallest interfacial energy is the most thermodynamically stable and energetically favored nucleation site for the heterogeneous nucleation of α-Al. As a result, it is highly possible that α-Al preferentially nucleates on the most close-packed plane of TiC_0.5_N_0.5_, i.e., (111)_TiCN_. The experimental results together with the DFT analysis provide the robust evidence that TiC_0.5_N_0.5_ can act as a potent nucleant for the α-Al grain to promote the grain refinement of α-Al.

It is noted that although NPs can be captured by the advancing solid/liquid interface at enhanced cooling conditions and thus distributed intragranularly throughout the matrix, they cannot be responsible for α-Al nucleation. In light of the classic nucleation theory, each nucleation site corresponds to one grain. As a result, merely a minority of NPs can participate in the nucleation events. Furthermore, the size of NPs and their size distribution have a significant influence on the nucleation efficiency. Based on the free growth theory [9], for a given particle size, the critical undercooling for grain initiation on the particle is fixed, above which the nuclei can continue to grow and below which the free growth of the crystal is stifled. At enhanced cooling conditions, the melt undercooling may be large enough to activate NPs as potent nucleants for the heterogeneous nucleation of α-Al, generating a high nucleation rate. However, a large undercooling can also promote the grain growth, which in turn gives a coarse grain size [17]. In fact, the nucleation of new crystals competes with the growth of nucleated grains [7,9,37]. The results of this work demonstrate that the effect of an enhanced nucleation rate outweighs that of fast grain growth. In addition, compared with the particles in grain refines, the added NPs have a relatively narrow size distribution, which favors intrinsically a fine grain size. As a result, it is inferred that the grain refinement at a high cooling rate is principally due to the NP-induced nucleation promotion.

On the basis of the analysis above, the grain refinement at different cooling rates can be qualitatively elucidated as follows. At relatively low cooling rates, the assembly of intergranular NPs onto the surface of growing crystals could stifle their growth. Meanwhile, a majority of NPs could not be activated as potent nucleants at very low undercoolings. As a consequence, the grain refinement at relatively low cooling rates may be due to the growth restriction caused by NPs. With increasing the cooling rates, the NP capture by SF proceeds and the growth restricting the effect of NPs is decreased. Additionally, the increased melt undercooling may reach the free-growth undercooling to activate NPs for grain initiation. Therefore, NP-induced heterogeneous nucleation could dominate in the grain refinement at relatively high cooling rates.

Apart from the microstructural refinement, according to our previous work [21,22,23], the TiCN NPs can impart high strength and ductility to the Al alloy. For one thing, the grain refinement induced by TiCN NPs can lead to the performance enhancement of materials, which is well described by classic Hall–Petch equation. For another, the ceramic nanoparticles distributed in the matrix, especially the intragranular ones at high cooling rates, could function as desirable reinforcements to immobilize dislocations via the Orowan strengthening and the coefficient of thermal expansion mismatch strengthening mechanisms, contributing to the improvement of material properties.

## 4. Conclusions

The addition of TiC_0.5_N_0.5_ nanoparticles can result in the grain refinement of α-Al at various cooling conditions. The microstructure analysis shows that the nanoparticles exhibit intergranular distribution and form an incoherent interface with α-Al at low cooling rates (<15 K/s), while they exhibit intragranular distribution and form the coherent interface with α-Al at high cooling rates (>70 K/s). The first-principles calculations reveal that among (111)_Al_//(111)_TiCN_, (200)_Al_//(200)_TiCN_, and (220)_Al_//(220)_TiCN_ interfaces, the (111) interface with the largest work of adhesion and smallest interfacial energy is the most thermodynamically stable and energetically favored nucleation site for the heterogeneous nucleation of α-Al. The experimental and theoretical results demonstrate that the NP-induced growth restriction and NP-induced nucleation promotion are responsible for the grain refinement at low and high cooling rates, respectively.

## Figures and Tables

**Figure 1 materials-16-01214-f001:**
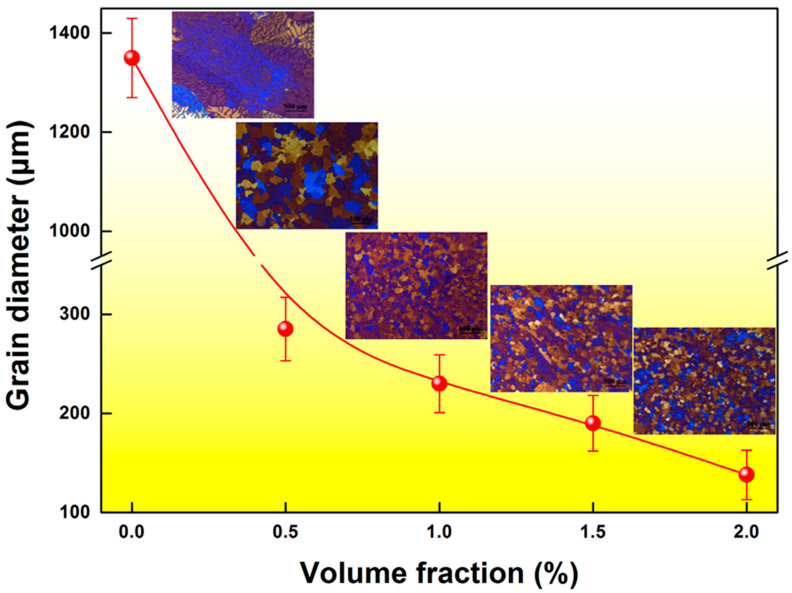
Variation in α-Al grain sizes with different NP addition levels. The insets are the anodized micrographs of the CP Al with various NP addition levels.

**Figure 2 materials-16-01214-f002:**
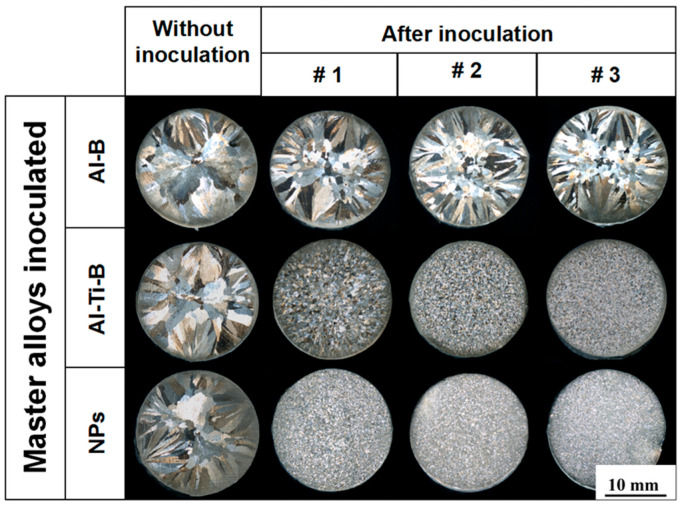
Comparison of NP-induced grain refinement with the conventional inoculation. For Al-B and Al-Ti-B master alloys, #1, #2, and #3 represent the addition level of 0.1 wt.%, 0.3 wt.%, and 0.5 wt.%, respectively. For NPs, #1, #2, and #3 represent the addition level of 0.5 vol.%, 1.0 vol.%, and 2.0 vol.%, respectively.

**Figure 3 materials-16-01214-f003:**
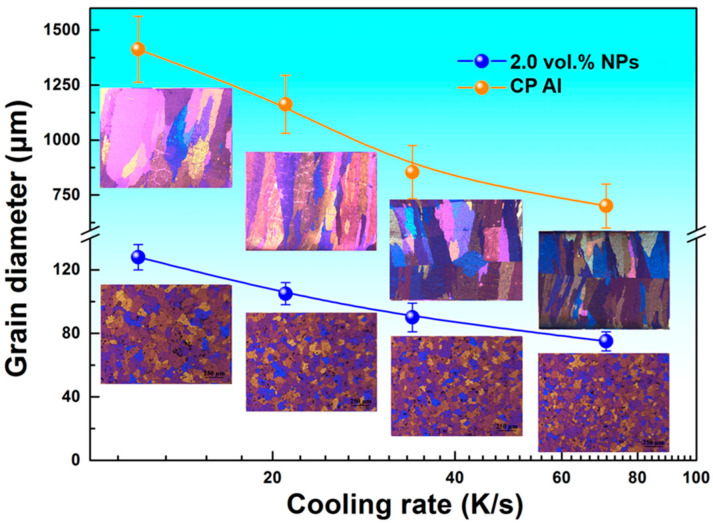
Variations in CP Al grain sizes with and without NP addition as functions of cooling rates. The insets are the anodized micrographs of the CP Al with and without NP addition solidified at different cooling conditions.

**Figure 4 materials-16-01214-f004:**
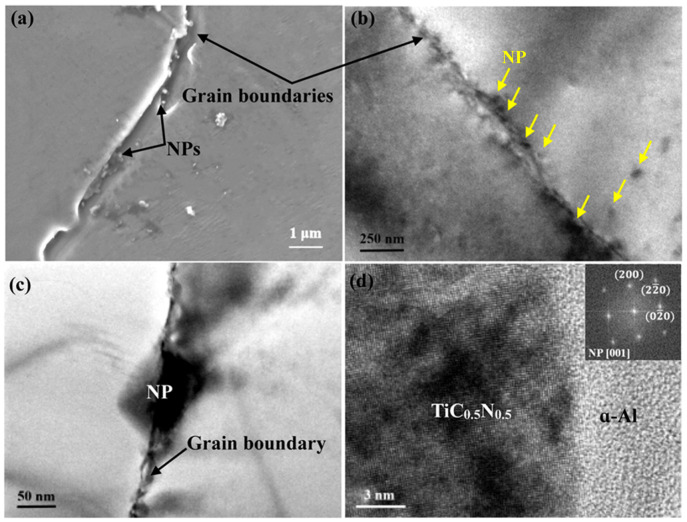
(**a**) SEM micrograph of CP Al with 2.0 vol.% NP addition solidified at the cooling rate of 10 K/s; (**b**,**c**) TEM bright field images showing the NP distribution along the grain boundary; (**d**) HRTEM image of the Al/TiC_0.5_N_0.5_ interface. The inset in (**d**) is the fast Fourier transformation (FFT) of the TiC_0.5_N_0.5_ nanoparticle.

**Figure 5 materials-16-01214-f005:**
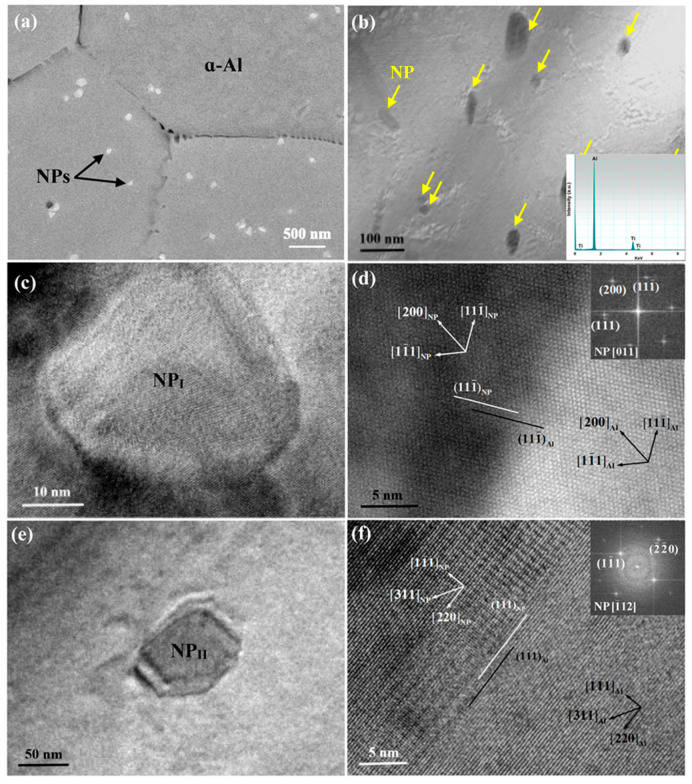
(**a**) SEM micrograph of CP Al with 2.0 vol.% NP addition solidified at the cooling rate of 70 K/s; (**b**) TEM bright field image showing the NP distribution within the interior of grain; (**c**,**e**) magnified view of the nanoparticle I (NP_I_) and nanoparticle II (NP_II_) in (**b**), respectively; (**d**,**f**) HRTEM images of the Al/NP_I_ and Al/NP_II_ interfaces, respectively. The insets in (**b**,**d**,**f**) are the EDS spectra of NPs and the fast Fourier transformation (FFT) of NP_I_ and NP_II_, respectively.

**Figure 6 materials-16-01214-f006:**
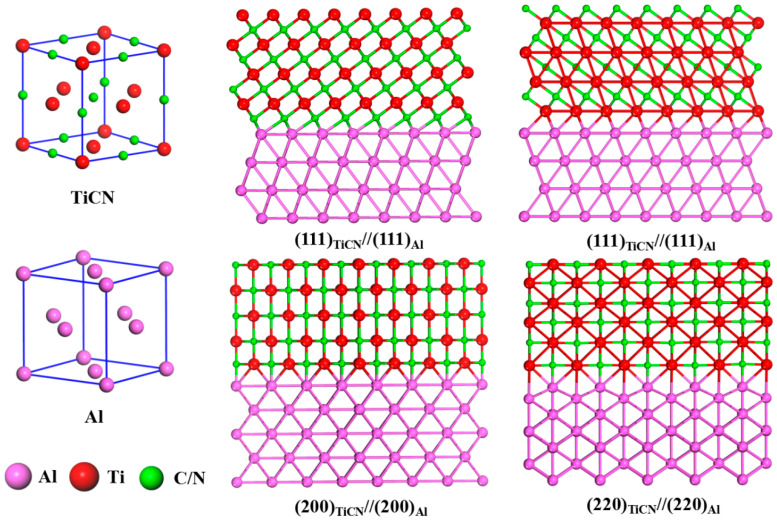
Interface structures of the Al matrix and TiCN nanoparticles.

**Figure 7 materials-16-01214-f007:**
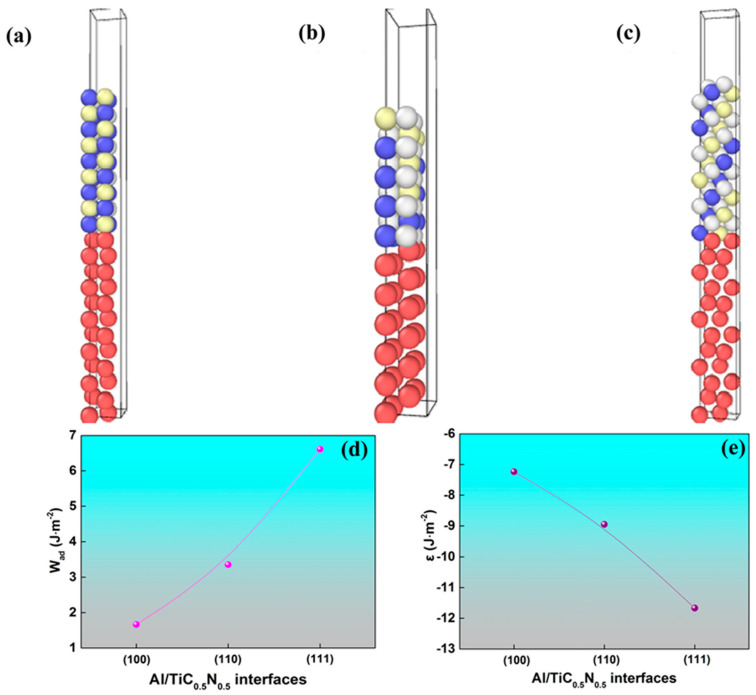
(**a**–**c**) Supercell models of (100), (110), and (111) Al/TiC_0.5_N_0.5_, respectively; (**d**) work of adhesion of three interface structures; (**e**) interfacial energy of three interface structures. Note that the red, blue, yellow, and white balls in (**a**–**c**) represent the Al, Ti, C, and N atoms, respectively.

**Table 1 materials-16-01214-t001:** Possible orientation relationships between TiC_0.5_N_0.5_ and Al.

Crystal Structure and Lattice Parameters (nm)	Matching Planes	*f_d_* (%)	Matching Direction	*f_r_* (%)	*OR*
N: Al FCC a = 0.4049 S: TiC_0.5_N_0.5_ FCC a = 0.4286	{111}*_N_*//{111}*_S_*	5.85	<110>*_N_*//<110>*_S_* <112>*_N_*//<112>*_S_*	5.86	{111}*_N_*//{111}*_S_*, <110>*_N_*//<110>*_S_*
					{111}*_N_*//{111}*_S_*, <112>*_N_*//<112>*_S_*
	{200}*_N_*//{200}*_S_*	5.84	<100>*_N_*//<100>*_S_* <110>*_N_*//<110>*_S_*	5.85	{200}*_N_*//{200}*_S_*, <100>*_N_*//<100>*_S_*
					{200}*_N_*//{200}*_S_*, <110>*_N_*//<110>*_S_*
	{220}*_N_*//{220}*_S_*	5.87	<100>*_N_*//<100>*_S_* <110>*_N_*//<110>*_S_* <112>*_N_*//<112>*_S_*	5.83	{220}*_N_*//{220}*_S_*, <100>*_N_*//<100>*_S_*
					{220}*_N_*//{220}*_S_*, <110>*_N_*//<110>*_S_*
					{220}*_N_*//{220}*_S_*, <112>*_N_*//<112>*_S_*

## Data Availability

All data included in this study are available upon request by contact with the corresponding author.

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
