# Peer review of "Grain Refinement Mechanisms of TiC0.5N0.5 Nanoparticles in Aluminum"

_materials, 2023, doi:10.3390/ma16031214_

Round 1

Reviewer 1 Report

kindly clarify the following

1. is DFT meaningful and predicted results at nano level.

2. are the crystal phases of all crystal interfaces transition 

3. can you predict Interface structures of the Al matrix and TiCN nanoparticles in crystal producer software suing CIF

4.could you make step by step NP-induced nucleation promotion at high cooling rates.

Author Response

Dear Editor and Reviewers,

Thank you very much for your and the reviewers’ comments on our manuscript entitled “Grain refinement mechanisms of TiC0.5N0.5 nanoparticles in aluminum” (materials-2156986). We have revised the manuscript carefully based on these comments. Revised portions have been highlighted by using yellow background. We have tried our best to improve the manuscript and made some changes in the manuscript. These changes will not influence the content and framework of the paper. We appreciate editors and reviewers’ work earnestly, and hope that the correction will meet with your approval.

Reviewer 1

Kindly clarify the following

  1. is DFT meaningful and predicted results at nano level.

Response: Thank you for your kind comments. As far as we know, the simulation at atomic/nano scale based on the DFT has been extensively applied and very meaningful for the scientific research.  

  1. are the crystal phases of all crystal interfaces transition

Response: Thank you for your kind comments. Based on our experimental results, not all crystal planes of α-Al have orientation relationships with TiCN, and (111)Al//(111)TiCN, (200)Al//(200)TiCN and (220)Al//(220)TiCN interfaces were found to be three best-matched interfaces between α-Al and TiCN.

  1. can you predict Interface structures of the Al matrix and TiCN nanoparticles in crystal producer software suing CIF

Response: Thank you for your valuable suggestion. At present, we have not used CIF to predict the interface structures of crystals, but we would try it in our future work.

  1. could you make step by step NP-induced nucleation promotion at high cooling rates.

Response: Thank you for your kind comments. According to the free-growth model proposed by Greer et al., the NPs can act as nucleation sites to promote the heterogeneous nucleation of  α-Al as soon as the increased melt undercooling could reach the free-growth undercooling.

Reviewer 2 Report

The author provides a manuscript on an interesting topic of Aluminium grain refinement using TiCN nanoparticle additives. Apparently, the article is a continuation of earlier studies conducted by the same group, which he and his colleagues first reported in 2016 (https://doi.org/10.1007/s11661-016-3668-3) and 2019 (https://doi.org/10.1016/j.ultsonch.2019.104626). In general, current article is well written and has some new results, namely Al-TiCN NP interface modelling), which were not reported in the previous studies. However, it is very clear that the experimental part of current study is presenting and discussing the properties of the same samples which were already comprehensively discussed and reported in the studies mentioned above. Author does a relatively good job and mostly avoids the direct plagiarism (with an exception of a debatable content of Fig.3), but at the same time he does not provide any reference to the earlier studies and this leaves a lot of questions unanswered, though they were discussed in the previous two studies. For example, earlier studies provided more structure data (including crystal structure measurement by XRD, elemental mapping etc.), more details on mechanical properties (for example nanoidentation hardness measurement details), etc.. Accordingly, in my opinion the current manuscript would be greatly improved if authors would included at least one or two paragraphs and provide a summary on how grain refinement by TiCN changes micro/macro properties of Al and provide reference to previous works.  Few more remarks and suggestions for corrections are provided bellow.

1.       At the top of the manuscript there is just one author, whereas at the left side there are few more names. Considering that this is a continuation report from previous studies, which had more authors, I would presume that the upper list is not full. This situation should be clarified, and roles of each contributing authors should be provided at the end of the manuscript.

2.       The possible plagiarism related to the content of Fig. 3 should be considered by authors/editors and if needed appropriate measures should be taken.

3.       Authors use a lot of abbreviations which are not explained in the text (OR, NPI, NPII, etc.). Abbreviation usage should be reviewed and full wording should be provided for each of them at its first usage.

4.       It is understandable, that the first column in Fig.2 presents images for the “pure” Al. But three identical images at first column for Al-B, Al-TiB and NPs still are confusing.  Authors are suggested to rethink image presentation to avoid such problem.  

5.       Fig. 7 should be reviewed to provide better proportions, larger axis labels, subscript for AD in Work of Adhesion.

6.       Introduction could be more informative and include some comments about the origins of such structure control method (for example authors can reference the work by J.Xu et al https://doi.org/10.1038/srep01730). Also it can be more specific about the advancement of current study results, namely the modelling part results.

Author Response

Dear Editor and Reviewers,

Thank you very much for your and the reviewers’ comments on our manuscript entitled “Grain refinement mechanisms of TiC0.5N0.5 nanoparticles in aluminum” (materials-2156986). We have revised the manuscript carefully based on these comments. Revised portions have been highlighted by using yellow background. We have tried our best to improve the manuscript and made some changes in the manuscript. These changes will not influence the content and framework of the paper. We appreciate editors and reviewers’ work earnestly, and hope that the correction will meet with your approval.

Reviewer 2

The author provides a manuscript on an interesting topic of Aluminium grain refinement using TiCN nanoparticle additives. Apparently, the article is a continuation of earlier studies conducted by the same group, which he and his colleagues first reported in 2016 (https://doi.org/10.1007/s11661-016-3668-3) and 2019 (https://doi.org/10.1016/j.ultsonch.2019.104626). In general, current article is well written and has some new results, namely Al-TiCN NP interface modelling), which were not reported in the previous studies. However, it is very clear that the experimental part of current study is presenting and discussing the properties of the same samples which were already comprehensively discussed and reported in the studies mentioned above. Author does a relatively good job and mostly avoids the direct plagiarism (with an exception of a debatable content of Fig.3), but at the same time he does not provide any reference to the earlier studies and this leaves a lot of questions unanswered, though they were discussed in the previous two studies. For example, earlier studies provided more structure data (including crystal structure measurement by XRD, elemental mapping etc.), more details on mechanical properties (for example nanoidentation hardness measurement details), etc.. Accordingly, in my opinion the current manuscript would be greatly improved if authors would included at least one or two paragraphs and provide a summary on how grain refinement by TiCN changes micro/macro properties of Al and provide reference to previous works. Few more remarks and suggestions for corrections are provided bellow.

  1.       At the top of the manuscript there is just one author, whereas at the left side there are few more names. Considering that this is a continuation report from previous studies, which had more authors, I would presume that the upper list is not full. This situation should be clarified, and roles of each contributing authors should be provided at the end of the manuscript.

Response: Thank you for your kind reminder. We have double-checked the author list and supplement the missing author names. Besides, the roles of each contributing authors have been provided at the end of the manuscript.

  1. The possible plagiarism related to the content of Fig. 3 should be considered by authors/editors and if needed appropriate measures should be taken.

Response: Thank you for your kind reminder. To avoid the possible duplication, we have replaced all the images in Fig.3 with the unpublished ones which were obtained in the same cooling rate areas.  

  1.       Authors use a lot of abbreviations which are not explained in the text (OR, NPI, NPII, etc.). Abbreviation usage should be reviewed and full wording should be provided for each of them at its first usage.

Response: Thank you for your kind reminder. According to the reviewer’s comment, we have scrutinized the manuscript and revised the abbreviation problem.

  1.       It is understandable, that the first column in Fig.2 presents images for the “pure” Al. But three identical images at first column for Al-B, Al-TiB and NPs still are confusing.  Authors are suggested to rethink image presentation to avoid such problem.  

Response: Thank you for your kind reminder. According to the reviewer’s suggestion, we have replaced the three images with different ones to avoid confusion.

  1. 7 should be reviewed to provide better proportions, larger axis labels, subscript for AD in Work of Adhesion.

Response: Thank you for your kind reminder. According to the reviewer’s suggestion, we have adjusted the Fig. 7 including enlarging axis labels and revising the subscript of Wad.

  1. Introduction could be more informative and include some comments about the origins of such structure control method (for example authors can reference the work by J.Xu et al https://doi.org/10.1038/srep01730). Also it can be more specific about the advancement of current study results, namely the modelling part results.

Response: Thank you for your kind reminder. According to the reviewer’s suggestion, we have revised Introduction and supplemented the comments about the work by J.Xu and the advancement of DFT.

“Most recently, the practice of incorporating nanoparticles (NPs) into the metallic melt has shown promising grain refinement results for Al and Mg alloys[18-20]. For instance, J.Xu et al have achieved the control of architecture and microstructure of Al composite by the assembly of NPs with liquid Al in molten salt[20]. ” 

“First-principles calculation based on density functional theory (DFT) is taken as a powerful and efficient tool for the prediction and estimation of the nucleation potency of nucleants. Up to now, little experimental work has been conducted to clarify the nucleation potency of NPs for α-Al, especially by the DFT calculation. ”

  1. Authors would included at least one or two paragraphs and provide a summary on how grain refinement by TiCN changes micro/macro properties of Al and provide reference to previous works.

Response: Thank you for your valuable suggestion. According to the reviewer’s suggestion, we have revised the manuscript and supplemented the discussion about how grain refinement by TiCN changes micro/macro properties of Al and provided reference to previous works.

“Apart from the microstructural refinement, according to our previous work[21-23], the TiCN NPs can impart high strength and ductility to the Al alloy. For one thing, the grain refinement induced by TiCN NPs can lead to the performance enhancement of materials, which is well described by classic Hall-Petch equation. For another, the TiCN nanoparticles distributed in the matrix, especially the intragranular ones at high cooling rates, could function as desirable reinforcements to immobilize dislocations via the Orowan strengthening and the coefficient of thermal expansion mismatch strengthening mechanisms, contributing to the improvement of material properties. ”

Reviewer 3 Report

The manuscript presents some interesting information on Grain refinement mechanisms of TiC0.5N0.5 nanoparticles in aluminum. I recommend that this manuscript should be subjected to a thorough grammar check and a great reduction in the percentage similarity index is highly needed for it to be publishable in MDPI. Should the authors be willing to address the following concerns;

Comment 1. In the introduction, the author mentioned “facilitating subsequent processing”. The statement should be written clearly.

Comment 2. The author made this statement in lines 61 to 65 “It is well accepted that a high imposed cooling rate can produce a large thermal undercooling at which NPs may be activated for heterogeneous nucleation. Hence, the objective of the present study is to clarify the combined effects of cooling rate and NP addition on grain refinement of aluminum”. There should be citations for this assertion.

Comment 3. The author mentioned this statement under the result and discussion section in lines 111-113 “macrostructure of the CP Al castings is characterized by coarse columnar grain structure. When the addition level is raised, a large number of equiaxed grains are formed at the center of the sample inoculated by Al-3B mater alloy”. What causes the coarse columnar grain structure after using an ultrasonicator for mixing?

Comment 4. In the result and discussion, lines 192 to 195, it seems the last sentence is incomplete. This should be revised to add meaning to the sentence.

Comment 5. In the same section in Figure 7. Where (a)-(c) Supercell models of (100), (110) and (111) Al/TiC0.5N0.5, are being mentioned. The authors need to unpack this more. The authors should put a reference on the equation raised since the claim made is not from their work.

Comment 6. In section 4.0, the conclusion statement is not clear. Please revise.

Comment 7. The plagiarism or the similarity index is not acceptable for publication in its current format. The authors should address it as soon as possible. 

Author Response

Dear Editor and Reviewers,

Thank you very much for your and the reviewers’ comments on our manuscript entitled “Grain refinement mechanisms of TiC0.5N0.5 nanoparticles in aluminum” (materials-2156986). We have revised the manuscript carefully based on these comments. Revised portions have been highlighted by using yellow background. We have tried our best to improve the manuscript and made some changes in the manuscript. These changes will not influence the content and framework of the paper. We appreciate editors and reviewers’ work earnestly, and hope that the correction will meet with your approval.

Reviewer 3

The manuscript presents some interesting information on Grain refinement mechanisms of TiC0.5N0.5 nanoparticles in aluminum. I recommend that this manuscript should be subjected to a thorough grammar check and a great reduction in the percentage similarity index is highly needed for it to be publishable in MDPI. Should the authors be willing to address the following concerns;

Comment 1. In the introduction, the author mentioned “facilitating subsequent processing”. The statement should be written clearly.

Response: Thank you for your kind comment. According to the reviewer’s suggestion, we have revised the manuscript.

“The microstructure control is of particular importance in the metal casting industry, and it is often the case that a uniformly fined, equiaxed grain structure is favorable for both cast and wrought alloys, as it benefits the casting process and improves the mechanical properties of the cast metal, facilitating subsequent mechanical working”

Comment 2. The author made this statement in lines 61 to 65 “It is well accepted that a high imposed cooling rate can produce a large thermal undercooling at which NPs may be activated for heterogeneous nucleation. Hence, the objective of the present study is to clarify the combined effects of cooling rate and NP addition on grain refinement of aluminum”. There should be citations for this assertion.

Response: Thank you for your kind comment. According to the reviewer’s suggestion, we have revised the manuscript.

“ It is well accepted that a high imposed cooling rate can produce a large thermal undercooling at which NPs may be activated for heterogeneous nucleation[9,16,17].”

Comment 3. The author mentioned this statement under the result and discussion section in lines 111-113 “macrostructure of the CP Al castings is characterized by coarse columnar grain structure. When the addition level is raised, a large number of equiaxed grains are formed at the center of the sample inoculated by Al-3B mater alloy”. What causes the coarse columnar grain structure after using an ultrasonicator for mixing?

Response: Thank you for your kind comment. In this work, the ultrasonic treatment (UT) was only exerted on the NP-containing samples and for the inoculation of Al-B and Al-Ti-B, no UT was applied. It is noted that because UT is applied above the liquidus of Al, UT is mainly effective in dispersing NPs. For the NP-free sample, the untreated microstructure of pure Al is is characterized by coarse columnar grain structure.

Comment 4. In the result and discussion, lines 192 to 195, it seems the last sentence is incomplete. This should be revised to add meaning to the sentence.

Response: Thank you for your kind reminder. The sentence is complete as it is “To better understand the effect of TiC0.5N0.5 NPs on the grain refinement of α-Al at enhanced cooling condition, their nucleation potency was examined                                                                                                                                                                                                                                                                                                                                                                                                                                                                                                                                                                                                                                                                                                                                                                                                                                                                                                                                                                                                                                                                                     in terms of the crystallographic matching, the work of adhesion and the interfacial energies of Al/TiC0.5N0.5 interfaces.”.

Comment 5. In the same section in Figure 7. Where (a)-(c) Supercell models of (100), (110) and (111) Al/TiC0.5N0.5, are being mentioned. The authors need to unpack this more. The authors should put a reference on the equation raised since the claim made is not from their work.

Response: Thank you for your kind comment. We have supplemented the discussion on the supercell models of (100), (110) and (111) Al/T

C0.5N0.5 and provided the citations for the equations.

“The supercell models of (100), (110) and (111) Al/TiC0.5N0.5 are displayed in Fig. 7(a)-(c), respectively. The models of polar Al2Ti2NC interface are the (1 1 1)Ti2NC || (1 1 1)Al, (1 1 0)Ti2NC || (1 1 0)Al, (1 0 0)Ti2NC || (1 0 0)Al, in which the close-packed planes are matched across the interface with N/C-terminated Ti2NC. To accommodate the periodic boundary condition inherent in a supercell calculation, we invoke the coherent interface approximation in which the softer Al matches the dimensions of the Ti2NC. To make the results comparable and reasonable, the ratio of between the content of C and that of N is adopted 1 on the interface for all Al/Ti2NC, and all the models are strictly stoichiometric, Al2Ti2NC. ”

Comment 6. In section 4.0, the conclusion statement is not clear. Please revise.

Response: Thank you for your kind comment. According to the reviewer’s suggestion, we have revised the conclusions.

“The addition of TiC0.5N0.5 nanoparticles can result in the grain refinement of α-Al at various cooling conditions. The microstructure analysis shows that the nanoparticles exhibit intergranular distribution and form the incoherent interface with α-Al at low cooling rates (<15 K/s), while they exhibit intragranular distribution and form the coherent interface with α-Al at high cooling rates (>70 K/s). The first-principles calculations reveal that among (111)Al//(111)TiCN, (200)Al//(200)TiCN and (220)Al//(220)TiCN interfaces, the (111) interface with the largest work of adhesion and smallest interfacial energy is the most thermodynamically stable and energetically favored nucleation site for the heterogeneous nucleation of α-Al. The experimental and theoretical results demonstrate that the NP-induced growth restriction and NP-induced nucleation promotion are responsible for the grain refinement at low and high cooling rates, respectively.” 

Comment 7. The plagiarism or the similarity index is not acceptable for publication in its current format. The authors should address it as soon as possible.

Response: Thank you for your kind comment. According to the reviewer’s suggestion, we have scrutinized the manuscript and revised it thoroughly.

Round 2

Reviewer 3 Report

The manuscript is good for publication in MDPI provided moderate English changes has been applied in some critical areas